# Biologically Potent Benzimidazole-Based-Substituted Benzaldehyde Derivatives as Potent Inhibitors for Alzheimer’s Disease along with Molecular Docking Study

**DOI:** 10.3390/ph16020208

**Published:** 2023-01-30

**Authors:** Bushra Adalat, Fazal Rahim, Wajid Rehman, Zarshad Ali, Liaqat Rasheed, Yousaf Khan, Thoraya A. Farghaly, Sulaiman Shams, Muhammad Taha, Abdul Wadood, Syed A. A. Shah, Magda H. Abdellatif

**Affiliations:** 1Department of Chemistry, Hazara University, Mansehra 21300, Pakistan; 2Department of Chemistry, COMSATS University, Islamabad 45550, Pakistan; 3Department of Chemistry, Faculty of Applied Science, Umm Al-Qura University, P.O. Box 715, Makkah Almukkarramah 24382, Saudi Arabia; 4Department of Biochemistry, Abdul Wali Khan University Mardan, Mardan 23200, Pakistan; 5Department of Clinical Pharmacy, Institute for Research and Medical Consultations (IRMC), Imam Abdul Rahman Bin Faisal University, P.O. Box 31441, Dammam 11099, Saudi Arabia; 6Faculty of Pharmacy, Universiti Teknologi MARA Cawangan Selangor Kampus Puncak Alam, Bandar, Puncak Alam 42300, Malaysia; 7Department of Chemistry, College of Sciences, Taif University, P.O. Box 11099, Taif 21944, Saudi Arabia

**Keywords:** benzimidazole, alzheimer disease, docking study, acetylcholinesterase and butyrylcholinesterase

## Abstract

Twenty-one analogs were synthesized based on benzimidazole, incorporating a substituted benzaldehyde moiety (**1**–**21**). These were then screened for their acetylcholinesterase and butyrylcholinesterase inhibition profiles. All the derivatives except **13**, **14**, and **20** showed various inhibitory potentials, ranging from IC_50_ values of 0.050 ± 0.001 µM to 25.30 ± 0.40 µM against acetylcholinesterase, and 0.080 ± 0.001 µM to 25.80 ± 0.40 µM against butyrylcholinesterase, when compared with the standard drug donepezil (0.016 ± 0.12 µM and 0.30 ± 0.010 µM, against acetylcholinesterase and butyrylcholinesterase, respectively). Compound **3** in both cases was found to be the most potent compound due to the presence of chloro groups at the **3** and **4** positions of the phenyl ring. A structure-activity relationship study was performed for all the analogs except **13**, **14**, and **20**, further, molecular dynamics simulations were performed for the top two compounds as well as the reference compound in a complex with acetylcholinesterase and butyrylcholinesterase. The molecular dynamics simulation analysis revealed that compound **3** formed the most stable complex with both acetylcholinesterase and butyrylcholinesterase, followed by compound **10**. As compared to the standard inhibitor donepezil both compounds revealed greater stabilities and higher binding affinities for both acetylcholinesterase and butyrylcholinesterase.

## 1. Introduction

Dementia is one of the major global challenges today. This dementia is mainly caused by Alzheimer’s disease (AD) [1]. The number of Alzheimer’s patients increases day by day as the population and age increase, by 2023 it will be up to 74.7 million people suffering from AD [2]. AD is a neurodegenerative disorder and reveals itself by the progressive loss of cognitive functions, speech impairment, and memory loss [3]. To understand the mechanism of action of the disease, several pathways have been identified. Neurochemically, it is characterized by the constant shortage in cholinergic neurotransmission that affects the cholinergic neurons in the basal forebrain [4]. This loss of cholinergic neurons results in the decrease of AchE in the cognition related areas of the brain, such as the cerebral cortex and the hippocampus [5]. AchE catalyzes the hydrolysis of acetylcholine into choline and acetic acid by two types of enzymes, acetylcholinesterase (AChE) and butyrylcholinesterase (BuChE) [6]. The brains of AD patients are rich in the AchE enzyme, which is responsible for the degradation of AchE [7]. Although at the cholinergic synapses the function of AchE is well known, the physiological function of BuChE is not familiar. However, it was observed that BuChE provides a backup to AChE when its activity decreases, and is active at high concentrations of AChE [8]. Another non-cholinergic role of AchE is the formation of senile plaque by the deposition of beta-amyloid (βA) [9]. In the beginning, therapeutic strategies for treating diminished cholinergic neurotransmission were mainly focused on AChE inhibitors, but studies have shown the significance of both AChE and BuChE [10]. Meanwhile, the inhibition of both these enzymes by a single molecule has been found to be a more important therapeutic agent for the treatment of AD. Cholinesterase inhibitors are classified into two classes: (i) specific, when they are used for the inhibition of AChE only; and (ii) nonspecific when they inhibit both AChE and BuChE. Among the drugs approved by the FDA, Tacrine and Rivastigmine are non-specific, while Donepezil and Galant-amine are specific, inhibiting AchE only [11]. However, these drugs have limited efficiencies and have various side effects, mostly at high doses.

Nitrogen containing heterocycles have received much attention in the field of drug discovery to cure various diseases, also various inhibitors are present in the literature to treat AD. Among these, benzimidazole is an important bioactive molecule, which is found naturally in Vitamin B12, and is widely used for the synthesis of novel drugs [12,13]. The heterocyclic ring of benzimidazole is also known as glyoxalin, 1,3-diazaole, imidazole, and imidazole is a term frequently used to show the five membered heterocyclic system containing tertiary N and imino functionalities in their structure. Imidazole rings are also found in naturally occurring compounds such as alpha-amino acids, and many proteins, such as purine, histamine, and biotin [14,15]. Benzimidazole scaffolds have been shown to have a varied range of biological activities such as antifungal [16,17], anticancer [18], antiulcer [19], antibacterial [20,21,22], anti-Alzheimer’s [23,24,25], and anti-inflammatory [26]. Various marketed drugs contain the benzimidazole moiety in their core structure; these include: Nocodazole (anticancer), Tiabendazole (antifungal and ant parasitic), Flubendazole (anthelmintic), and omeprazole (proton pump inhibitor). 

Our research group is continuously trying to introduce new heterocyclic compounds containing different moieties, for better therapeutic agency. Recently, we have reported the benzimidazole bearing moiety in the skeleton for an α-glucosidase inhibitor [27,28], an anti-Alzheimer’s inhibitor [29], and an *alpha*-glycosidase inhibitor [30,31,32,33], but still there is a need to discover more compounds for future research. So, we identified interesting profiles of substituted benzimidazole derivatives and screened them for anti-Alzheimer’s activity as shown in Figure 1.

## 2. Results and Discussion

### 2.1. Chemistry

Twenty-one scaffolds of 4-methoxybenzene-1,2-diamine and various substituted benzaldehyde based benzimidazoles were synthesized. First of all, 4-methoxybenzene-1,2-diamine (I, 0.5 mmol) with substituted benzaldehyde (0.5 mmol) in DMF (10 mL), in the presence of the catalyst sodium meta-bisulfate (0.5 mmol), and the resulting mixture was refluxed for 2–3 h to obtain the targeted derivatives (**1**–**21**) with the appropriate yield. Thin Layer Chromatography (TLC) was employed for the monitoring of the reaction till the conformation. Different spectroscopic techniques such as 13CNMR, 1HNMR, and HR-EIMS were carried out to confirm the structure of the synthesized analogs (Figure 1).

### 2.2. Molecular Docking Studies

We conducted a molecular docking analysis to understand the binding interactions of compounds with active site residues of the selected enzyme. Based on the co-crystal of the crystallographic structures, the synthesized compounds were docked into the active sites of specific enzymes. The native inhibitor from the AChE was removed and then re-docked into the active site using MOE. This approach was used to validate the docking procedure [34]. Utilizing PyMOL 2.3, the re-docked complex was then superimposed on the reference co-crystallized ligand to calculate the root mean square deviation (RMSD) which was predicted to be 0.56, revealing the validity of the docking protocol. The native inhibitor was removed from the AChE and then re-docked according to the docking technique, every compound was given a total of twenty conformations. The most active compounds’ top-ranked conformations were chosen for future studies and visual inferences. The docking results revealed that compound **3** showed excellent inhibitory potential against AChE against both targets. In the case of the binding mode of this compound, we have found that several key residues adopted key interactions with the essential compound moiety as showed in Figure 2a. The detailed interaction profile revealed residues Tyr 334 (via Pi-stacking) with the 5-ring. Also, we have found that this residue was further stabilized by the residue Asp72, because of the proximity, and having Van der Walls interactions between them. Several other Pi-stacking interactions were observed which might have a key role in the stability of this compound in the active site of AChE, i.e., Ph. residues around this moiety. The high potential might be due to the attached electron withdrawing groups at the meta and para positions of the benzene. These groups withdraw some of the electrons from the Pi-system, and subsequently, create a partial positive charge over the benzene ring, and next this ring is compelled to adopt several Pi-stacking interactions with key residues. Comparing this result with other similar compounds in the series also showed potential against the target enzyme. Compounds **10** and **11** showed similar behaviours; these compounds have similarly attached electron withdrawing groups at different positions. The only differences are the attached hydroxyl group at the ortho position and the attached Cl position. The lower potential compared to potent compounds might be due to the attached OH group, which is categorized as an electron donating group. The potential of this compound might be due to the Cl group withdrawing electrons from the Pi-system, and ultimately the donating group donates electrons to stabilize the overall system, Figure 2b,c. On the other hand, compound **3** showed high potential against BuChE. The mechanism of inhibition might be due the reason discussed in the above section, Figure 2a. The second potent compound in the series, 11, showed a good potential compared with the least active compound. Though the differences found among the most active and second ranked compounds is just the variation in the attached groups, which ultimately affects the activity of the overall compound. The protein-ligand interaction profiles of other active compounds are shown in Figure 2b,c. The computational and bioassay studies showed that these compounds are the most active and significant compounds, which showed the best potential against both the targets. The drug likenesses of the most active compounds are presented in Table 1.

### 2.3. Post Simulation Analysis

#### 2.3.1. RMSD (Root Mean Square Deviation) Analysis

In order to assess the stability of the systems, the root mean square deviation (RMSD) was calculated for each complex. Figure 3 displays the RMSD curve for the two best docking score compounds **3** and **10**, and a reference, in complex with AChE. All systems reported an RMSD that ranged from 1 to 2 Å. The RMSD of compound **3** in complex with AChE revealed a minor fluctuation between 5–10 ns and 20–25 ns, after that the system converged and remained stable until the end of the 50 ns MD simulation. Compound **10** in complex with AChE revealed fluctuations between 5–10 and 20–30 ns, but soon attained equilibrium and remained stable during the whole 50 ns MD run. The reference compound donepezil in complex with AChE was stable during the first 10 ns, after that the system showed major fluctuations up to 30 ns. During this period, the RMSD increased to 2 Å. After that, the system converged and attained stability and remained stable up to 50 ns MD. Further, the RMSD analyses of compounds **3** and **10,** and the reference, in complex with BuChE are presented in Figure 4. The RMSD analyses demonstrated that the compounds were stable, and fit into the binding pockets of AChE and BuChE. The compound **3** complex exhibits the lowest RMSD among all the complexes, as shown in Figure 4. As compared to the standard inhibitor, compounds **3** and **10** showed a greater stability for both of the targets.

#### 2.3.2. RMSF (Root Mean Square Fluctuation) Analysis

To take into account changes in the amino acid residue, the RMSF of the AChE and BuChE backbone residues were calculated. To further understand how ligand binding impacts the flexibility of each residue during the simulation, the RMSF was investigated. The stability, stiffness, and compactness of the receptors were indicated by the amino acid residues with the lowest RMSF values. Figure 5 shows the estimated RMSF values for selected compounds and for the AChE complexes. Residues including Val 280, Leu 281, Val 282, Asn 283, His284, Glu285, Trp286, His287, Val288, Leu289, Pro290, Ser 399, Trp 500, Pro 501, and Pro 502 indicated a high degree of fluctuation, while the active site residues such as Tyr 334, Asp72, Phe 288, Phe 331, Tyr 121 indicated great stability. Figure 5 displays RMSF plots for compounds **3**, **10,** and the reference, in complex with AChE, while Figure 6 displays the RMSF plots for compounds **3**, **10,** and the reference, in complex with BuChE. 

### 2.4. Binding Energy Calculation

Numerous techniques have been employed for virtual screening, docking, molecular dynamics, and MMPBSA free energy calculations of compounds. In this study, the MMPBSA.py python script was used for calculating the binding energy [35]. The binding energy of compound **3** in complex with AChE was found to be −56 kcal/mole, while compound **3** in complex with BuChE was found to be −37 kcal/mole. The binding energies of compound **10** in complex with AChE and BuChE were predicted to be −52 kcal/mole and −31 kcal/mole, respectively. The delta G value of the best-docked complexes was good as compared to the standard inhibitor donepezil. The results are shown in Table 2. 

### 2.5. Pharmacokinetics (ADMET) Properties of Finally Selected Compounds

Traditional drug design and discovery is a dangerous investment, which is commonly exposed to unpredicted failures in various stages of the drug discovery and development. One main reason for these failures is the efficiency and safety faults, which are related largely to absorption, distribution, metabolism, excretion (ADME) properties, and different toxicities (T). Therefore, rapid ADMET analysis is urgently needed to reduce the chance of failure in the drug discovery process. pkCSM is an online server that conveniently performs six types of drug-likeness analysis (five rules of Lipinski and one prediction model), 31 ADMET endpoints prediction includes three basic properties, six absorptions, three distributions, ten metabolisms, two excretions, and seven toxicities. pkCSM is a free online server accessible at http://biosig.unimelb.edu.au/pkcsm/prediction. 

ADMET properties have been studied for the three best compounds finally selected, i.e., compounds **3**, **10**, and **11**, along with the reference compound, having an effective IC_50_ value in vitro, with the best docking scores. All these compounds obeyed Lipinski’s rule of five, according to which, “a drug like compound must not have more than 10 hydrogen bond acceptors, not more than 5 hydrogen bond donors, a octanol-water coefficient not more than 5, and the molecular weight must be less than 500 Daltons”. They also had ADMET properties in the required allotted range, which guarantees their drug likeness. Assays were carried out for compounds **3**, **10**, and **11**, and their ADMET properties are shown in Table 3 and Table 4.

### 2.6. Structure-Activity Relationship (SAR)

We have synthesized twenty-one scaffolds of substituted benzimidazole that exhibit varying degrees of cholinesterase inhibition potential when compared with the standard drug Donepezil, having IC_50_ values of 0.016 ± 0.12 and 0.30 ± 0.010 for acetylcholinesterase and butyrylcholinesterase, respectively. The potent compounds among the series were compounds **1**–**2**, **4**, **6**–**12**, **15**–**16**, and **9**–**21**, having IC_50_ values ranging from 0.050 ± 0.001 to 5.80 ± 0.10 for acetylcholinesterase, and 0.080 ± 0.001 to 5.90 ± 0.10 for butyrylcholinesterase. The most potent compounds against AChE were, compound **3** (IC_50_ = 0.050 ± 0.001, 0.080 ± 0.001), which had two chloro groups on the phenyl ring at positions 3 and 4, and compound **10** (IC_50_ = 0.10 ± 0.001), which had a hydroxyl group at position 2 and chloro groups at positions 3 and 5. The most potent compound against BuChE was compound **11** (IC_50_ = 0.30 ± 0.001), which had chloro groups at positions 2 and 4, shown in Figure 7. This shows that the position of substituents will affect the inhibition potential of analogs.

Scaffolds **15** and **16** both have hydroxyl groups, but their positions are different. In compound **15** the hydroxyl group is at the para position, and it has IC_50_ values of 0.80 ± 0.001 and 0.90 ± 0.001 for AChE and BuChE, respectively. While for analog **16**, the hydroxyl group is at the meta position, and the IC_50_ values of this compound are 1.30 ± 0.10 and 2.10 ± 0.10 for AChE and BuChE, respectively (Figure 8). This shows that the ideal position for the hydroxyl group is para.

Comparison of the nitro substituted compounds, **1** (IC_50_ = 2.10 ± 0.10, 1.60 ± 0.10), **2** (IC_50_ = 5.10 ± 0.10, 5.90 ± 0.10), and **4** (IC_50_ = 3.40 ± 0.10, 3.60 ± 0.10) shows that analog **1**, in which the nitro group is at the para position, is superior in activity to analogs **2** and **4**, which have the nitro group at the ortho and meta positions. Thus, positional changes of the nitrogen atom causes a decline in the activity in the ortho and meta analogs (Figure 9).

Similarly, if we compare analog **9** (2.30 ± 0.10, 2.10 ± 0.10), having one hydroxyl and one methoxy group, with analog 21 (2.10 ± 0.10, 2.70 ± 0.10), having one hydroxyl and two methoxy groups, analog 21 showed a better inhibition potential than analog **9**. Analog **7**, having a benzoyloxy group on the phenyl ring also exhibited good inhibition, with an IC_50_ value of 5.80 ± 0.10 for AChE, and 5.90 ± 0.10 for BuChE (Figure 10). While analog **6** (IC_50_ = 4.10 ± 0.10, 5.10 ± 0.20), having a methyl group, which is an electron donating group, also showed potent inhibition, but less than analogs **3**, **10**, and **11**. The SAR study shows that the presence of electron withdrawing groups or electron donating groups on the phenyl ring play a crucial role in the inhibition profile. The rest of the series of compounds exhibited poor inhibition.

## 3. Materials and Methods

### 3.1. Procedure for the Synthesis of Benzimidazole Analogs (***1***–***21***)

First of all, 4-methoxybenzene-1,2-diamine (I, 0.5 mmol), with variously substituted benzaldehyde (0.5 mmol) in DMF (10 mL) in the presence of the catalyst sodium meta-bisulfate (0.5 mmol), and the resulting mixture was refluxed for 2–3 h to obtain the targeted derivatives (**1**–**21**) with appropriate yield. Thin Layer Chromatography (TLC) was employed for the monitoring of the reaction until the desired conformation. At the end, the final product was separated, washed with distilled water, and then dried.

#### 3.1.1. 5-Methoxy-2-(4-Nitrophenyl)-1H-Benzo[d]-Imidazole (**1**)

1HNMR (500 MHz, DMSO-d6): δ 11.31 (s, 1H, NH), 8.14 (d, J = 7.4 Hz, 2H, Ar-H), 8.10 (d, J = 7.9 Hz, 1H, Benzimidazole-H), 7.84 (s, 1H, Benzimidazole-H), 7.52 (d, J = 6.8 Hz, 1H, Benzimidazole-H), 7.40 (d, J = 7.3 Hz, 2H, Ar-H), 2.33 (s, 3H, -OCH3), 13CNMR (125 MHz, DMSO-d6): δ 168.4, 163, 160.3, 149.7, 147.8, 143.3, 140.4, 126.1, 123.6, 121.3, 118.2, 117.7, 113.9, 45.6. HR-EIMS: m/z Calcd. for C14H11N3O3 [M] + 269.0800; Found. 269.0796.

#### 3.1.2. 5-Methoxy-2-(3-Nitrophenyl)-1H-Benzo[d]-Imidazole (**2**)

1HNMR (500 MHz, DMSO-d6): δ 11.36 (s, 1H, NH), 8.18 (dd, J = 7.2, 2.0 Hz, 1H, Ar-H), 8.12 (d, J = 6.9 Hz, 1H, Benzimidazole-H), 7.87 (s, 1H, Benzimidazole-H), 7.58 (d, J = 6.4 Hz, 1H, Benzimidazole-H), 7.24 (s, 1H, Ar-H), 7.20 (t, J = 6.5 Hz, 1H, Ar-H), 7.06 (dd, J = 7.2, 1.9 Hz, 1H, Ar-H), 2.38 (s, 3H, -OCH3), 13CNMR (125 MHz, DMSO-d6): δ 168.5, 160.5, 149.6, 147.9, 143.1, 140.0, 134.7, 126.2, 123.7, 121.4, 118.4, 117.8, 113.0, 45.7. HR-EIMS: m/z Calcd. for C14H11N3O3 [M] + 269.0800; Found. 269.0796.

Sample analysis can be found in Appendix A.

### 3.2. Inhibition Assay Protocol of Acetylcholinesterase and Butyrylcholinesterase

The AChE/(BuChE) inhibitory test was measured using a spectrophotometric method developed by Elman et al. [31,32]. A buffer solution containing phosphate (pH 8.0) of volume 140 μL; 20 μL of each AChE/BuChE solution; and 20 μL of the test sample were incubated at room temperature for 15 min. AChE/BuChE 10 μL was used to start the reaction, followed by the addition of DTNB. ATCh or BTCh hydrolyzed the reaction of DTNB with thiocholine for 15 min; unrestricted by AChE and BuChE enzymatic hydrolysis. E − S/E × 100, where E&S represent enzyme activity with and without test samples, were used to compute the percentage (percent) inhibition (30). Each sample’s inhibitory activity was measured in terms of IC_50_ (g/mL) or μM. For all substances, the IC_50_ values were derived using a generic graph. The graph was created in Excel and the IC_50_ values were derived by taking Y = 50 and determining the x value as IC_50_.

### 3.3. Molecular Docking

A molecular docking conformation study was carried out to understand the basics of the binding modes of the synthesized compounds against the selected enzymes, acetylcholinesterase (AChE) and butrylcholinesterase (BuChE), in order to corroborate the in vitro and in silico results using the Molecular Operating Environment (MOE) software package. Both targets’ crystal structures were extracted from the protein data bank (RCSB) by using codes (PDB) 1ACL for AChE and 1P0P for BuChE. While the proteins and all synthesized compounds were protonated and the energy was minimized by using the MOE Dock modules default parameters, resulting in optimal structures for both the proteins and the compounds. Docking investigations were conducted using these optimized structures of the target proteins and compounds. Our earlier investigations have detailed descriptions of the protocol [33,34].

### 3.4. Molecular Dynamics simulation

To confirm the stability of the docked complexes, a systematic molecular dynamics simulation was run using the AMBER20 software. The TIP3P water model was used to dissolve each system in a rectangular box, and the systems were then neutralized by introducing counter ions [35]. Steepest descent minimization for 6000 cycles and conjugate gradient minimization for 3000 cycles were used to minimize the energy of all the neutralized systems. The systems were quickly heated to 300 K after the completion of the energy minimization. Then, each system underwent a two-step equilibration process at a constant temperature of 300 K and 1 atm. The density was equilibrated for 2 ns in the first stage using a weak constraint. The systems were then allowed to stabilize for more than 2 ns without any constraints in the second step. The production phase was then conducted for 50 ns. The particle mesh Ewald technique was employed for long-range electrostatic interactions with a cutoff distance of 10.0 [36]. Covalent bonds were calculated using the SHAKE algorithm [37,38]. The cpptraj package was used to evaluate trajectory data while molecular dynamic simulations were carried out using the GPU-supported pmemd.cuda [39,40].

## 4. Conclusions

In conclusion, we have designed and synthesized a range of benzimidazole base derivatives (**1**–**21**), and all of the synthesized derivatives were assessed for inhibition of acetylcholinesterase and butyrylcholinesterase. All the synthetic analogs showed different values of IC_50_, ranging from 0.050 ± 0.001 to 25.30 ± 0.40 against acetylcholinesterase, and from 0.080 ± 0.001 to 25.80 ± 0.40 against butyrylcholinesterase, as compared with the standard drug donepezil (0.016 ± 0.12 µM against acetylcholinesterase, and 0.30 ± 0.010 µM against butyrylcholinesterase). Compounds **13**, **14**, and **20** did not show any activity. Compound **3** in both cases showed excellent inhibitory potential due to the presence of chloro groups at the **3** and **4** positions of the phenyl ring, and a structure-activity relationship study was carried out for all the analogs except **13**, **14**, and **20**. All the newly synthetic compounds were characterized by various spectroscopic techniques to confirm the structures, such as 1HNMR, 13CNMR, and HR-EIMS. A molecular docking study was performed to understand the binding interactions between the analogs and proteins. It is clear from the results that benzimidazole derivatives in the present work could be considered as active neuroprotective therapeutics in the future.

## Data Availability

Data is contained within the article and Appendix A.

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
