# Peer review of "Biologically Potent Benzimidazole-Based-Substituted Benzaldehyde Derivatives as Potent Inhibitors for Alzheimer’s Disease along with Molecular Docking Study"

_pharmaceuticals, 2023, doi:10.3390/ph16020208_

Round 1

Reviewer 1 Report

The report for the manuscript ID pharmaceuticals-2093252-peer-review-v1.pdf “Biologically Potent Benzimidazole-based-substituted benzaldehyde Derivatives as Potent Inhibitors for Alzheimer’s Disease along with Molecular docking study”. In the following, I have included only a few comments that support the shortcomings of this manuscript.

Comment 1. Please pay attention to spelling details, e.g.:

line 5 – the author’s names: Bushra Adalata,..? or Bushra Adalat? Etc

lines 10 and 15 – the letters “c” and “f” must be removed

line 26 - against acetylcholinesterase ? or against butyrylcholinesterase. The same observation for lines 229-230

lines 24-29  - are copy-paste in lines 228-232

line 102 - phi-stacking?  Or Pi-stacking (line 130)

lines 176-196 – in subsection 3.1 only synthesis of only two compounds are presented

lines 197- 1. nhibition….? Or Inhibition?

etc.

Comment 2. The “Introduction” section of the manuscript requires extensive revision.  I would suggest that the authors attempt to present the key objectives of their study with regards to what is currently known (i.e. literature), thus highlighting the added value of the paper. The introduction is very small. Please increase its size and underline the impact of your work. More profound discussions and comparisons with other published works are welcomed.

Comment 3. Why did the authors choose the 1ACL and 1P0P for docking? In PDB there are many structures of Human Acetylcholinesterase in Complex with Donepezil (the reference drug for the manuscript).

Comment 4. The docking procedure developed by the authors is very poorly presented. A simple validation of the docking procedure is required. Please provide the computed RMSD value between the docked pose and the cocrystallized ligand.

Comment 5. Please provide a higher quality image of Figure 2. Without a clear representation of the ligand bound to the binding site, it is very hard to imagine what is going on. Maybe the figure could be completed with a 2D representation of interactions.

Comment 6. I strongly recommend binding free energy calculations using MM/PBSA and/or MM/GBSA because they are more accurate than most scoring functions of molecular docking or, if possible, at least 100 ns MD simulation to establish the stability followed by post-simulation binding free energy.

Comment 7. Druglikeness criteria and ADME predictions analysis could be an additional argument for selected compounds to be considered potential AD inhibitors. I recommend to compute these parameters for the prioritized compounds.

Comment 8. The Conclusion section could be improved by advantageously presenting the work and the obtained results.

Reviewer hopes that the comments are useful and allow authors to improve the manuscript quality, and also to rewrite the paper to bring it closer to the standards of the journal Pharmaceuticals. The manuscript should be considered for publication in Pharmaceuticals following major revision.

Author Response

Reviewer 1

Comment 1. Please pay attention to spelling details, e.g.:

line 5 – the author’s names: Bushra Adalata,..? or Bushra Adalat? Etc

Authors response: According to the kind reviewer’s suggestion the authors names corrected Bushra Adalat1, Fazal Rahim1, Wajid Rehman*1, Zarshad Ali1, Muhammad Taha2, Liaqat Rasheed1, Thoraya A. Farghaly3, Sulaiman Shams4, Abdul Wadood4, Syed A. A. Shah5, Magda H. Abdellatif6

lines 10 and 15 – the letters “c” and “f” must be removed

Authors response: According to the kind reviewer’s suggestion the letters “c” and “f” removed.

line 26 - against acetylcholinesterase ? or against butyrylcholinesterase. The same observation for lines 229-230

Authors response: According to the kind suggestion of reviewers is now corrected.

lines 24-29  - are copy-paste in lines 228-232

Authors response: According to kind suggestion of reviewers is now corrected.

line 102 - phi-stacking?  Or Pi-stacking (line 130)

Authors response: According to kind suggestion of reviewers is Pi-stacking now corrected.

lines 176-196 – in subsection 3.1 only synthesis of only two compounds are presented

Authors response:

lines 197- 1. nhibition….? Or Inhibition? etc.

Authors response: According to kind suggestion of reviewers is now corrected is Inhibition.

Comment 2. The “Introduction” section of the manuscript requires extensive revision.  I would suggest that the authors attempt to present the key objectives of their study with regards to what is currently known (i.e. literature), thus highlighting the added value of the paper. The introduction is very small. Please increase its size and underline the impact of your work. More profound discussions and comparisons with other published works are welcomed.

Authors response: As per kind suggestions of reviewers suggested corrections are incorporated.

Comment 3. Why did the authors choose the 1ACL and 1P0P for docking? In PDB there are many structures of Human Acetylcholinesterase in Complex with Donepezil (the reference drug for the manuscript).

Presently, we are not familiar with the 1ACL and 1P0P for docking? In PDB there are many structures of Human Acetylcholinesterase in Complex with Donepezil (the reference drug for the manuscript, as our group is already use to and have access 1ACL and 1P0P for docking, however we appreciate the reviewer’s suggestions and will be used for in our upcoming papers after getting’s well know how of the suggested.

Comment 4. The docking procedure developed by the authors is very poorly presented. A simple validation of the docking procedure is required. Please provide the computed RMSD value between the docked pose and the cocrystallized ligand.

Author response: According to the kind reviewer’s suggestion the docking validation is now added to the manuscript as highlighted in the revised manuscript.

Comment 5. Please provide a higher quality image of Figure 2. Without a clear representation of the ligand bound to the binding site, it is very hard to imagine what is going on. Maybe the figure could be completed with a 2D representation of interactions.

Author response: The quality of Figure 2 is now improved.

Comment 6. I strongly recommend binding free energy calculations using MM/PBSA and/or MM/GBSA because they are more accurate than most scoring functions of molecular docking or, if possible, at least 100 ns MD simulation to establish the stability followed by post-simulation binding free energy.

Author response: According to the reviewer’s suggestion MD simulation and binding energy calculation is now added to the manuscript as highlighted in the revised manuscript.

Comment 7. Druglikeness criteria and ADME predictions analysis could be an additional argument for selected compounds to be considered potential AD inhibitors. I recommend to compute these parameters for the prioritized compounds.

Author response: Drug likeness for selected compounds is now added.

Comment 8. The Conclusion section could be improved by advantageously presenting the work and the obtained results.

Authors response: According to the kind suggestion of reviewer The Conclusion now improved.

Reviewer 2 Report

We found the manuscript relevant to the field, containing interesting information about the benzimidazole-derivatives synthesized molecules targeting AChE and BChE inhibition.

However, some deep concerns were raised about enzyme kinetics methodology and the manuscript result’s reproducibility based on the details given in the methods section. The experimental design regarding inhibition kinetics lacks adequate depth (non-linear regression estimations) to test the hypothesis (the new drug has an advantage over known drugs). So, the in vitro methodology is the core issue to test the hypothesis under study, along with molecular docking studies to support it. In our opinion, an inappropriate study of enzyme inhibition, at the early stages of drug development, can make it more expensive and delay the process of the "new lead" process.

A thorough and very careful review of citations and references should be done! We found repeated references (e.g. 12 and 13) and some not suitable to the material developed in the text (e.g. 2,7, 29-31,....).

Specific Comments/Suggestions:

#1 _L21-22_ The presented values refer to which inhibition parameter (IC50, Ki....)?

#2_ L28-33_ Rewrite, please, for clarity of content. Also, the term “excellent inhibition potential” (L28) must be supported by appropriate enzyme kinetic studies along with the molecular docking studies done.

#3_ Figure 2_ Edit, please, to increase quality, they appear distorted.

#4_ In the Material and Methods, we didn’t find any Statistical analysis methodology to support the estimates of Table 1. To be noticed that “mean±SD” and “mean±SEM” are not the same. How was the Standard Error estimated?

#5_ L197-208 The cited references appear to be either a review on AD pathology [29] or methodologies related to a-glucosidase kinetics [30-31]. 

Author Response

Reviewer 2

However, some deep concerns were raised about enzyme kinetics methodology and the manuscript result’s reproducibility based on the details given in the methods section. The experimental design regarding inhibition kinetics lacks adequate depth (non-linear regression estimations) to test the hypothesis (the new drug has an advantage over known drugs). So, the in vitro methodology is the core issue to test the hypothesis under study, along with molecular docking studies to support it. In our opinion, an inappropriate study of enzyme inhibition, at the early stages of drug development, can make it more expensive and delay the process of the "new lead" process.

A thorough and very careful review of citations and references should be done! We found repeated references (e.g. 12 and 13) and some not suitable to the material developed in the text (e.g. 2,7, 29-31,....).

Authors response: As per kind suggestions of reviewers suggested corrections are incorporated same references are replaced.

Specific Comments/Suggestions:

#1 _L21-22_ The presented values refer to which inhibition parameter (IC50, Ki....)?

According to the kind suggestion of reviewer the presented values refer to IC50 inhibition parameter.

#2_ L28-33_ Rewrite, please, for clarity of content. Also, the term “excellent inhibition potential” (L28) must be supported by appropriate enzyme kinetic studies along with the molecular docking studies done.

According to the kind suggestion of reviewer are incorporated.

#3_ Figure 2_ Edit, please, to increase quality, they appear distorted.

According to the kind suggestion of reviewer quality of picture increased now.

#4_ In the Material and Methods, we didn’t find any Statistical analysis methodology to support the estimates of Table 1. To be noticed that “mean±SD” and “mean±SEM” are not the same. How was the Standard Error estimated?

According to the kind suggestion of reviewer the Standard Error estimated through “mean±SEM” in Table-3 after revision at page no. 8.

#5_ L197-208 The cited references appear to be either a review on AD pathology [29] or methodologies related to a-glucosidase kinetics [30-31]. 

The cited reference covers the mentioned work in general

Round 2

Reviewer 1 Report

The report for the manuscript ID pharmaceuticals-2093252-peer-review-v2.pdf “Biologically Potent Benzimidazole-based-substituted benzaldehyde Derivatives as Potent Inhibitors for Alzheimer’s Disease along with Molecular docking study”.

The authors have tried to correct the highlighted questions but they omitted some of them. Please pay close attention to all questions/recommendations. For example: for comment 1, both in the manuscript and in the cover letter, I found only partial answers from the authors. The same observation for comment 7. Additionally, the author’s answer to comment 3 is not clear.

My opinion is the same, although interesting, the manuscript needs major revision.

Author Response

Comment 1. Please pay attention to spelling details, e.g.:

line 5 – the author’s names: Bushra Adalata,..? or Bushra Adalat? Etc

line 26 - against acetylcholinesterase ? or against butyrylcholinesterase. The same observation for lines 229-230

lines 24-29  - are copy-paste in lines 228-232

lines 176-196 – in subsection 3.1 only synthesis of only two compounds are presented

lines 197- 1. nhibition….? Or Inhibition? etc.

The spelling and grammar errors have been checked thoroughly and have been corrected

.

.

Comment 3. Why did the authors choose the 1ACL and 1P0P for docking? In PDB there are many structures of Human Acetylcholinesterase in Complex with Donepezil (the reference drug for the manuscript).

Presently, we are not familiar with the 1ACL and 1P0P for docking? In PDB there are many structures of Human Acetylcholinesterase in Complex with Donepezil (the reference drug for the manuscript, as our group is already use to and have access 1ACL and 1P0P for docking, however we appreciate the reviewer’s suggestions and will be used for in our upcoming papers after getting’s well know how of the suggested.

Comment 7. Druglikeness criteria and ADME predictions analysis could be an additional argument for selected compounds to be considered potential AD inhibitors. I recommend to compute these parameters for the prioritized compounds.

Author response: Drug likeness for selected compounds is now added.; in addition ADMET calculations in details have been calculated and incorporated

Reviewer 2 Report

We appreciate the authors´ effort in their responses.

The manuscript has been improved on most of the suggestions made. However, we maintain our opinion that the widespread use of IC50, which is devoid of any mechanistic foundation, must be replaced by the appropriated inhibition constants.

IC50 corresponds to the concentration of inhibitor that slows the rate (v0) of an enzyme-catalyzed reaction, in the absence of an inhibitor, to half that initial rate. Under specific conditions of ionic strength, temperature and pH, at change with the inhibitor dissociation constant, IC50 is not a constant.

This parameter (IC50), widely used in the literature is extremely incorrect when the actual enzyme kinetic mechanism, towards a given inhibitor, is non-linear e.g. partial and hyperbolic inhibitions (real mechanism associated with the most potent inhibitors!!). This fact has been the subject of concern by senior enzymologists (and not only) for decades. As an indication, Professor Antonio Baici in Chapter 4 of " Kinetics of Enzyme-Modifier Interactions. 2015 " [DOI:https://doi.org/10.1007/978-3-7091-1402-5] presents a treasured discussion on this subject, offering solutions for complex mechanisms (partial and hyperbolic inhibitions).

We hope that in the future, the dear authors will reconsider the oversimplified use of this parameter, devoid of mechanistic foundation, to the detriment of the appropriate use of the kinetic inhibition constants of the enzyme's real mechanism. In our opinion, the Compound 3 of this manuscript is worth it.

Author Response

The manuscript has been improved on most of the suggestions made. However, we maintain our opinion that the widespread use of IC50, which is devoid of any mechanistic foundation, must be replaced by the appropriated inhibition constants.

IC50 corresponds to the concentration of inhibitor that slows the rate (v0) of an enzyme-catalyzed reaction, in the absence of an inhibitor, to half that initial rate. Under specific conditions of ionic strength, temperature and pH, at change with the inhibitor dissociation constant, IC50 is not a constant.

This parameter (IC50), widely used in the literature is extremely incorrect when the actual enzyme kinetic mechanism, towards a given inhibitor, is non-linear e.g. partial and hyperbolic inhibitions (real mechanism associated with the most potent inhibitors!!). This fact has been the subject of concern by senior enzymologists (and not only) for decades. As an indication, Professor Antonio Baici in Chapter 4 of " Kinetics of Enzyme-Modifier Interactions. 2015 " [DOI:https://doi.org/10.1007/978-3-7091-1402-5] presents a treasured discussion on this subject, offering solutions for complex mechanisms (partial and hyperbolic inhibitions).

We hope that in the future, the dear authors will reconsider the oversimplified use of this parameter, devoid of mechanistic foundation, to the detriment of the appropriate use of the kinetic inhibition constants of the enzyme's real mechanism. In our opinion, the Compound 3 of this manuscript is worth it.

Reply: We are very much indebted for the encouraging comments of the worthy reviewer, though most of the literature reported in the field is based on IC 50 values, however the reviewer suggestion would be taken into consideration and we will try to work hard on the suggested parameters. I once again thankful to worthy reviewer for consideration of our manuscript

Round 3

Reviewer 1 Report

The report for the manuscript ID pharmaceuticals-2093252-peer-review-v3.pdf “Biologically Potent Benzimidazole-based-substituted benzaldehyde Derivatives as Potent Inhibitors for Alzheimer’s Disease along with Molecular docking study”.

The authors addressed the main concerns from the reviews.

I have the following two remarks on the content:

1. Supplementary material must be mentioned in the main text.

2. The Supplementary material is not presented according to the requirements of the Pharmaceuticals journal.

Author Response

Supplementary material must be mentioned in the main text.

Reply: Supplementary materials is now mentioned in the main text of the manuscript

  1. The Supplementary material is not presented according to the requirements of the Pharmaceuticals journal.

Reply: Incorporated as per requirements
